# Occupational Safety and Health Equity Impacts of Artificial Intelligence: A Scoping Review

**DOI:** 10.3390/ijerph20136221

**Published:** 2023-06-24

**Authors:** Elizabeth Fisher, Michael A. Flynn, Preethi Pratap, Jay A. Vietas

**Affiliations:** 1Division of Environmental and Occupational Health Sciences, School of Public Health, University of Illinois Chicago, Chicago, IL 60612, USA; fishere@uic.edu (E.F.); plakshmi@uic.edu (P.P.); 2Division of Science Integration, National Institute for Occupational Safety and Health, Cincinnati, OH 45226, USA; dse4@cdc.gov

**Keywords:** artificial intelligence, future of work, algorithmic bias, algorithmic integrity, health equity, occupational safety and health

## Abstract

Artificial intelligence (AI) has the potential to either reduce or exacerbate occupational safety and health (OSH) inequities in the workplace, and its impact will be mediated by numerous factors. This paper anticipates challenges to ensuring that the OSH benefits of technological advances are equitably distributed among social groups, industries, job arrangements, and geographical regions. A scoping review was completed to summarize the recent literature on AI’s role in promoting OSH equity. The scoping review was designed around three concepts: artificial intelligence, OSH, and health equity. Scoping results revealed 113 articles relevant for inclusion. The ways in which AI presents barriers and facilitators to OSH equity are outlined along with priority focus areas and best practices in reducing OSH disparities and knowledge gaps. The scoping review uncovered priority focus areas. In conclusion, AI’s role in OSH equity is vastly understudied. An urgent need exists for multidisciplinary research that addresses where and how AI is being adopted and evaluated and how its use is affecting OSH across industries, wage categories, and sociodemographic groups. OSH professionals can play a significant role in identifying strategies that ensure the benefits of AI in promoting workforce health and wellbeing are equitably distributed.

## 1. Introduction

Artificial intelligence (AI) is at the core of the development of “Cyber-Physical Systems” that characterize the current paradigm shift in the world of work known as the Fourth Industrial Revolution [1]. As such, understanding the impact of AI on the health, safety, and wellbeing of workers is essential to the field of occupational safety and health [2]. While not uniformly defined, artificial intelligence (AI) refers to systems built to execute intellectual processes of humans such as reasoning, identifying meaning, generalizing information, or learning from experience. Employers use numerous applications of AI to streamline business processes and increase worker productivity and safety. Algorithms that make recruiting and hiring decisions are increasingly common ways to identify the fit of job candidate. These algorithms use natural language processing to extract information from resumes to create a database of potential hires [3,4]. Algorithms that rely on facial recognition are used to screen video interviews for a job candidate’s body language, speech cadence, and communication skills [5]. Human resource conversational bots and smart assistants manage onboarding and responding to employee questions, thereby reducing the workload for strained human resource professionals [6]. AI allows for real-time data collection to monitor workers and identify exposure risk. [7]. Environmental sensors and biosensors collect data on biological, physical, or chemical changes and convert them into measurable signals that flag early warning signs of occupational disease or distress [8]. Robotics in manufacturing, healthcare, transportation, and logistics, among other industries, rely on AI and sensor technologies to remove human contact from dangerous or risky work tasks [9]. While there is evidence describing the potential of AI to increase productivity and protect the health and safety of workers in different industry settings [10], the ways in which AI either reduces or exacerbates occupational health inequities in the workplace must be explored.

Occupational safety and health (OSH) inequities are avoidable differences in work-related injury and illness incidence, morbidity, and mortality that are closely linked with social, economic, and environmental disadvantage resulting from structural and historical discrimination or exclusion [11]. Social and economic structures can lead to occupational health inequities in a variety of ways, including the overrepresentation of workers from certain social groups in dangerous occupations [12,13], language barriers [14], differential treatment on the job [15], and limited access to resources, including technologies, that help protect workers on the job [16,17]. For example, research suggests that wages, education, race, and place of birth are all associated with OSH outcomes such as employment in high injury/illness occupations, mortality [10], and job insecurity [13]. In addition, U.S.-based studies suggest that workers who are from certain racialized ethnic minority groups, such as Black, American Indian/Alaskan Native, and Hispanic, have a high school degree or less, or are foreign-born are at disproportionate risk for negative OSH outcomes [10,18].

### The Artificial Intelligence and Occupational Safety and Health Equity Research Gap

The same social structures that contribute to occupational health inequities influence the development, distribution, and integration of new technologies at work [14]. Understanding and accounting for the influence these exclusionary social structures play in the development and implementation of new technologies is essential in order to reduce their likelihood of aggravating existing occupational health inequities and unlock their potential to mitigate them [14]. The growing presence of AI at work, along with changing workforce demographics, shifting work arrangements, the digital skills gap, and increasing technological job displacement, makes understanding the relationship between AI and occupational health inequities a central question in Future of Work (FoW) research [19]. The FoW initiative was established by the U.S. National Institute for Occupational Safety and Health (NIOSH) in 2019 to prepare OSH professionals to address future workplace exposures and hazards. A research agenda for the FoW initiative was published in 2021, identifying a need to further explore the equitable distribution of technology-related OSH risks and benefits (Goal 7, Objectives 1–2) [20]. This study begins to address the research gap by exploring the role of AI in reducing or exacerbating occupational health inequities within the context of the FoW. A scoping review of the literature, thematic representation of the findings, and discussion of considerations for OSH professionals to address AI and OSH equity are included in this review.

## 2. Materials and Methods

In this study, a scoping literature review was completed to summarize the recent literature on AI’s role in OSH equity. A scoping literature review aims to synthesize the research in a topic area in order to serve as a starting point for future research [21]. Scoping provides more flexibility than a systematic review or meta-analysis, as it accounts for more diverse and relevant literature (e.g., gray literature) [22]. This literature review was guided by two priori assumptions: (1) AI likely contributes to OSH inequities; and (2) OSH professionals may not be fully prepared to address the health and wellbeing impacts of AI in workplaces. The following research questions (RQs) guided the analysis of full-text literature.
RQ 1: How can AI be used to promote OSH equity?RQ 2: How does AI present barriers or challenges to OSH equity?RQ 3: What are best practices for addressing emerging OSH equity challenges related to AI?
What are gaps in addressing AI and OSH equity?What are key issues for OSH/Industrial Hygiene professionals to understand around AI?


The delimiting of a literature search topic ensures that the search remains manageable. At the onset of the literature search, keywords and concepts were identified as they relate to an AI system’s ability to promote health equity [23,24]. The keywords and concepts related to AI, OSH, and equity presented in Table 1 were identified to guide the search frame.

The scoping review search strategy is demonstrated in Figure 1 using an adapted PRISMA methodology [28]. This strategy was used to retrieve peer-reviewed, white and gray literature, industry-specific papers, think tank reports, and other related publications. Publications used in the scoping literature review were assessed for eligibility and inclusion. The search strategy was designed around the aforementioned keywords/concepts.

Through an iterative process and as themes were identified in the literature, the initial search terms were amended and additional relevant literature was included [23]. An iterative process was used to ensure that we captured as much of the relevant literature as possible during the identification step of our search. Keywords describing these concepts were searched for in online databases and on the websites of the Top 50 think tanks as represented by the University of Pennsylvania Think Tank Index Report [29]. The Think Tank Index Report is identified through an international survey of approximately 2000 scholars, funders, policy makers, and journalists, who rank over 6500 think tanks using evidence-informed criteria developed by the University of Pennsylvania Think Tanks and Civil Societies Program. Direct searches on the websites of U.S.-based government agencies were completed. Government agencies were identified by two study team members based on existing records of AI use and/or research related to the NIOSH FoW initiative. Accordingly, governmental literature reflects the U.S. landscape. All search sources are listed in Table 2. When available, controlled vocabulary (e.g., Medical Subject Headings) was employed. Search terms (Table 3) were used to search through article content for inclusion. Key events were searched for upon identification in the literature. An example of a key event is the failure of the Maneuvering Characteristic Augmentation System (MCAS) of Boeing 737 Max airplanes, resulting in two crashes. The MCAS AI system was designed to activate and assist a pilot under certain circumstances; however, technical issues resulted in fatal accidents [30]. Our inclusion criteria required publication during 2005–2021 to account for advances in AI and in AI’s role in the workplace. Publication type (i.e., empirical, commentaries, etc.) was not limited in the scoping review. Peer-reviewed, white, and grey literature publications were translated into English. Duplicate publications were excluded.

The initial scoping results revealed 468 articles, with 135 articles screened on abstract. Full text screening included 135 articles: 79 peer-reviewed journal articles, 32 pieces of white literature, and 24 pieces of gray literature. Full-text screening identified 75 articles for inclusion: 31 peer-reviewed journal articles, 29 pieces of white literature, and 15 pieces of gray literature. Secondary citations were searched based on in-text references to the previously identified keywords/concepts. These 38 secondary citations were included in the scoping review, and did not need to meet all inclusion criteria (e.g., date of publication). In total, 113 articles were analyzed (n = 113): 49 peer-reviewed journal articles, 39 from the white literature, and 25 from the gray literature. A single reviewer identified the articles for inclusion, and the study team was consulted at each stage of the search. If a keyword and/or concept was referenced at least one time in the article, its use was recorded in an Excel spreadsheet. Author, year, title, source type (e.g., peer review), knowledge type (e.g., empirical), methods, key points, and pertinent quotes were collected, and the ways in which the article answered the RQs were recorded. Appendix A includes a table of the 113 articles included in the scoping review analysis.

## 3. Results

Of the 468 articles initially identified in the scoping review, 75 articles were selected for inclusion. Through full-text screening of the 75 articles, an additional 38 articles were identified as secondary citations. The 38 secondary citations were included in the analysis for a total sample of 113 articles (see Appendix A). The 113 articles were analyzed for reference to the keywords and concepts used to guide the search (see Table 1). The keyword and concept most frequently referenced in the reviewed literature was job security (n = 58), while confidentiality (n = 39) was the least frequently mentioned. The frequency of each keyword/concept is presented in Table 4. Human-centered system design (n = 48) is essential to building trusted systems that will not create or exacerbate OSH concerns caused by AI, and may even ameliorate them; this includes creating explainable AI. Explainable AI, where users are able to understand the operations, the outputs of the system, and how they are used, can improve trust and efficiency. This reduces the “black box” problem, which obscures the inner workings of AI systems, causing distrust [31,32,33]. The concept of bias was mentioned in 35% (n = 40) of the screened full-text articles. AI systems, notably algorithms, collect, maintain, analyze, and share protected worker information.

Job security was added as a fifth conceptual domain prior to full-text screening because it was frequently noted in the literature. The literature suggests that AI affects job security because it may be an existential threat to segments of the workforce, possibly resulting in mass unemployment in certain industries and occupations [34,35,36,37]. In spite of this, research suggests that the advent of new technologies may merely transition occupations and create new jobs, with only temporary job insecurity [38,39,40,41,42,43]. Privacy (n = 43) remains a critical issue in AI implementation and use, and is discussed further below [44,45,46,47,48,49].

The results of the literature review demonstrate certainty that AI has a significant role in the FoW. However, the role of AI is divergent, as it can both facilitate OSH equity and create barriers to it. For example, AI used in factories and warehouses may use “machine vision” to reduce the risk of robot–human collision; however these AI systems often fail to recognize darker skin tones, which increases injury/fatality risk for workers who are Black, Indigenous, or people of color (BIPOC) [43,50].

In the course of our review, three significant themes emerged around AI’s ability to promote and present barriers to OSH equity (RQs 1&2): (1) AI’s Impact on High-Risk Industries; (2) Data Use and Algorithmic Integrity; and (3) Societal Shifts. Best practices to address emerging challenges in OSH equity (RQ 3) are described as well. A summative interpretation of the literature, recommendations for future research, and considerations for OSH professionals are described in the discussion section.

### 3.1. Promoting OSH Equity: AI’s Impact on High-Risk Industries and Precarious Work

In response to RQ 1, AI may be both a barrier and facilitator to OSH. AI has significant potential to improve OSH equity, particularly in high-risk industries such as construction, manufacturing, mining, and oil and gas transportation. AI may reduce the need for workers to engage in dirty, dangerous, or monotonous work, or to work in extreme conditions such as poor weather and emergency or disaster situations [31,51,52,53]. These jobs are more likely to be occupied by workers that are from racialized ethnic minority groups, have a high school degree or less, are foreign-born, and receive low wages [13]. Traumatic occupational injuries are geographically clustered; clusters of occupational injuries are correlated directly with immigrant communities and urban poverty in the U.S. [54]. Indicative of health inequities and OSH disparities, certain communities take on a higher burden of dangerous work and traumatic injuries. By reducing exposure to hazardous conditions in these industries, AI has the potential to reduce occupational health inequities for workers from these communities.

AI technologies may enhance OSH for workers in high-risk industries through various safety optimization systems. AI’s ability to collect real-time exposure data may improve exposure estimates, predict adverse events, and reduce the impact of hazards [31,38,55,56,57,58]. Examples of such AI technologies include operator alert systems, remote imaging technology, and use of environmental sensors and biosensors to measure exposure levels [59,60,61]. However, reliance on AI systems presents concerns for workers as well. Certain AI applications, such as productivity trackers that monitor activity, present increased instances of surveillance and control for worker populations already experiencing lower levels of socioeconomic status (SES) [62,63]. In the FoW, the potential misuse of workforce data is likely to increase as AI integration becomes ubiquitous [64,65]. For example, jobs that require GPS tracking (e.g., platform-based rideshare drivers) may reveal protected personal information about an employee (i.e., sexuality or religion) that can be used to discriminate against them [66]. Low-wage workers, workers without collective bargaining units, temporary workers, and workers in precarious jobs may be more likely to work in industries that use wearable data collection devices, which may track workflow efficiency or be used to monitor wellness initiatives [67]. Thus, due to the nature of their work arrangements, these workers may be more likely to have their data exposed and may be susceptible to security violations. This requires algorithmic integrity in the form of proper systems to curb the mishandling and misuse of received data in order to reduce bias [48,68].

### 3.2. Barriers and Facilitators to OSH Equity: Data Use and Algorithmic Integrity

Scheduling and hiring AI systems are notably discussed in the literature as ways to potentially reduce gender, racial, ability, and age bias in the workplace. However, previous research has demonstrated that AI hiring tools are more susceptible to inherent bias and discrimination than anticipated [32,69,70]. AI has been shown to automatically discover hidden patterns in natural language datasets, leading systems to capture patterns that reflect human biases such as racism, sexism, ageism, and ableism [71]. To the degree that AI reduces the consideration of these patterns in the decision-making process, it could help reduce bias and inequities in hiring and scheduling at work. Alternately, recognition of these patterns could reinforce or amplify existing biases present in society if these patterns influence the AI decision-making process. While the effects of algorithmic bias are not yet delineated in the literature, there is reason for concern. The ability to recognize patterns in human diversity is one way AI can reinforce existing bias and occupational health inequities. For example, speech recognition AI, which has been used in hiring, has demonstrated clear biases against African Americans and groups with dialectical and regional speech variations [44].

Conversely, machine learning algorithms that do not account for human diversity can reinforce existing bias and occupational health inequities as well. For example, AI facial analysis has shown clear disparities across skin color, failing to detect darker skin tones, and is highly concerning for people with disabilities due to failure to recognize craniofacial differences or mobility devices [8,43,72]. Certain algorithms have misidentified darker-skinned women as often as 35% of the time and darker-skinned men 12% of the time, which is much higher than the same rates for Caucasians [73,74]. As such, any benefits from this AI facial analysis would disproportionally favor able-bodied individuals with lighter skin tones, thereby reinforcing existing social inequities.

### 3.3. Barriers and Facilitators to OSH Equity: Societal Shifts

AI implementation at work will result in societal shifts related to job security that will be influenced by existing social inequalities related to unemployment, digital divides, and skill gaps. These societal shifts are likely to disproportionally impact workers with lower SES, thereby creating additional barriers to OSH equity and aggravating existing social inequalities that are detrimental to health. Early in the scoping review search process, job security emerged as a conceptual domain due to its recurrence in the literature.

Workers are at risk of being outperformed by AI [75]. AI has the potential to cause the loss of jobs in certain industries. As much as 40%–50% of the workforce, or more in developing countries, is vulnerable to technological job displacement [8,76]. In care roles, for example, research suggests that the elderly may prefer robots to humans for certain tasks [77,78,79].

Numerous predictions have been offered on how AI will impact unemployment, with the most frequently noted being that AI may cause short-term instability and job losses in certain sectors while creating jobs in others [33,36,80,81,82]. Others predict that AI will affect both lower-skilled workers (e.g., drivers, security, and cleaners) and highly skilled workers (e.g., lawyers, physicians, analysts, and managers) [67,83]. Certain sectors, such as trucking, will be particularly hard hit; other such sectors include the service industry and healthcare. However, AI’s impact on unemployment may be exaggerated, and the successful retooling and reskilling of labor may mitigate job loss [37,39]. AI’s use across industries, workplaces, and society requires consideration of which social groups may have limited opportunities to build the skills necessary to succeed in the FoW.

While automation may increase productivity and create new jobs, these new jobs might not be equitably distributed across racial groups, genders, age groups, U.S. geographical regions, or skill levels [32,40,84,85]. Those with higher education or better access to job training and workforce development programs are more likely to succeed in the FoW [83,86]. Employers are likely to experience immediate skills gaps, and highly skilled workers with knowledge of AI will be more valued and employable than those without AI skills [86,87].

The digital divide will likely increase existing education, skill [34,88,89,90,91], OSH, and income inequities [45,73,82,92]. The digital divide and technological displacement may be most apparent in historically marginalized communities, for example, in rural and low-income communities. These communities will experience the most significant disruptions to job security (the same American communities most negatively impacted by IT-era changes) [84,93]. Digitally-oriented metropolitan areas with large populations that are better educated will experience less disruption from AI integration. Employees of small businesses (i.e., businesses employing fewer than eleven individuals) may be less likely to have their OSH concerns reported to state and federal agencies [94]; therefore, the impact of AI on these workers is likely to be understudied. Existing AI systems have only been evaluated at a small number of worksites with limited geographic diversity within the U.S., which reduces generalizability and lessons learned for use in rural, low-income communities, small businesses, or low-resourced organizations [95,96,97].

The digital divide is compounded in historically marginalized communities by existing differences in school resources that correlate with residential and income segregation in low-income communities. High-resourced schools are better poised to prepare students for changes to the labor market and provide them with the skills necessary to successfully adapt to technological advances at work such as the growing reliance on AI [89]. Wealth inequality may escalate as AI investors and workers from high-resourced communities subsume the majority share of income growth [98]. Conversely, technological job displacement, at least initially, will disproportionally impact workers from low-resourced communities, leading to increased job insecurity. Perceived job insecurity and anxiety over new skills has been likened to a public health crisis [99]. Increases in depression, suicide, and alcohol and drug abuse, including opioid-related death, may occur, especially among individuals from low-resourced communities that already experience health inequities [31,100,101,102].

### 3.4. Best Practices for Emerging Challenges

The literature included in the scoping review often provided recommendations for successful AI implementation, which have been summarized below to suggest best practices for emerging challenges related to AI. Accordingly, companies that develop and use AI must be aware of the ethical, legal, and societal impacts of AI integration in the workplace [41,98,103]. The literature suggests that it is possible to design AI in a way that promotes equity and reduces biases using an iterative risk informed approach to research, design, and implementation [104]. Human-centered systems consider how humans will interface with the AI, which reduces biases and places realistic demands on workers. Consequently, there is less likelihood of health and safety concerns in such a situation because the AI is developed with consideration of specific cultural, economic, and social environments [92].

An ethical code or framework for justice in AI development and implementation was frequently recommended in the literature, and would facilitate OSH equity [74,105]. Ethicists [43,69] or third-party auditors have been recommended to ensure appropriate use of AI [36,42,106]. Algorithmic audits assess a system for bias, accuracy, robustness, interpretability, privacy characteristics, and other unintended consequences [34]. In order to reduce bias, the diversification of algorithm training data and ensuring that training data mirrors the demographic diversity of the population with respect to which the algorithm is used are fundamental to equity [46,65,107]. There is an existing need to diversify the AI workforce (i.e., developers, coders/programmers) and provide equitable opportunities for employment to ensure more representative AI [108]. Providing education on AI development to BIPOC while engaging workers in the development of AI through participatory approaches to AI design and implementation may facilitate the use of more diverse algorithmic training data.

In light of the impending skills gap, opportunities for AI skills training [105,109], retooling of labor [34,87], and continuing educational opportunities that are accessible to all workers [83,110], especially those from historically underserved communities, are recommended to promote OSH equity. “Future-proofing” workers includes training that builds soft skills, foundational skills, and technical/occupational skills related to the FoW [84]. Support in finding roles that complement AI systems or enable transitioning to new jobs as needed may assist those subject to job displacement [90,111]. Pathways to emerging skills, such as computer operation technology, software development technology, mechanical manufacturing, automatic control, and intelligent control, can be developed; apprenticeships and alternative learning models may be especially effective for adult learners [112,113]. Education systems can be adapted to ensure that all communities are prepared for the future of work and that all students have the digital skills necessary to succeed in the workforce. Inequitable opportunities for education as a result of racial or gender bias reinforce and perpetuate health inequities by limiting access to job opportunities and healthcare [89,114].

Certain communities, particularly those that are smaller, low-income, rural, or historically underserved, will need support in order to adjust to the potential negative impacts of AI [85]. Serious economic and labor market disruption can be mitigated by intentional community reinforcements. Universal basic income [34,45,47] and increased social safety nets are considered viable options for reducing income inequality or the impacts of job displacement [112,115]. Universal internet access can increase equity; it has been suggested that telecommunication firms, internet providers, and satellite companies be incentivized in order to expand and improve their networks in underserved communities [74,88,116].

## 4. Discussion

This scoping review has identified ways in which AI promotes OSH equity. AI tools such as biosensors [8,10,31,72] and wearable technologies [48,73,102], can reduce the impact of workplace hazards through continuous monitoring of workers’ chemical, physical, biological, and ergonomic risk while on the job. AI integration in the workplace can improve OSH outcomes, particularly in high-risk industries [14,87,92]. Algorithmic recruiting, hiring, and scheduling all have the potential to reduce historical or systemic biases, although AI hiring tools are more susceptible to inherent bias and discrimination than anticipated [44,67,107]. It is essential to incorporate inclusive research and design practices when developing and implementing AI applications for the workplace [117]. Additional research on how human-centered systems design is operationalized to improve worker safety may further justify its use in the development of AI.

Facilitators of OSH equity include efforts to involve employees in decision-making around AI [118], hiring corporate ethicists or conducting audits of AI systems [37,43,46,96], and using representative algorithm training data [106,107]. Social safety nets such as universal basic income [45,47] and universal internet access [92,113] may improve equity in communities that experience negative impacts of AI integration.

One barrier to OSH equity is that reliance on certain AI systems creates privacy and confidentiality concerns and increased worker surveillance that can disadvantage workers if safeguards on use of the data generated by these systems are not implemented [62,63]. Privacy and confidentiality concerns can be mitigated by increased transparency of power structures, algorithmic audits, and multidisciplinary participatory approaches to AI design, implementation, maintenance, and evaluation.

Concerns around job security [34,75], the inequitable distribution of new jobs [45,83], and increased income disparities [88,89,92] associated with AI are well-founded and associated with negative health effects [100,101]. The use of AI in the workplace may further disadvantage populations of the American workforce that already experience structural vulnerabilities, such as women, BIPOC persons, rural workers, and those experiencing job precarity or displacement [14,116].

As technology advances, a risk that the OSH benefits will not be equitably distributed exists, which may aggravate existing occupational health inequities or create new ones. OSH professionals must understand and account for the societal shifts caused by AI in the workplace as they develop programs to promote the safety, health, and wellbeing of their workers. AI may cause short-term instability and job losses in certain industry sectors and demographic groups while creating jobs in others [33,80,81]. Robust and applied measures of the current and future effects of AI job displacement and job creation are currently in development [31]. Economists assert that the impact of new technologies is conceptualized as enabling labor to be more productive if technology implementation is gradual [87]. Mechanisms to facilitate the rapid retooling of labor in pace with AI development and expansion are necessary in order to reduce OSH inequities.

### 4.1. Recommendations for Future Research

This paper begins to identify knowledge gaps on AI and OSH equity that can inform FoW research. Knowledge gaps include limited research on the practical effects of existing AI used in workplaces. This includes algorithmic data representation to reduce bias as well as how AI systems are evaluated for their impact on the safety, health, and wellbeing of workers [43,44,75]. The methods used to evaluate and explain the accuracy of AI, along with the metrics, standards, and guidelines for the ethical use of AI, are lacking as well. Future research is needed to explore issues relating to technological job displacement and mitigating the digital divide. In addition, research is needed that considers where AI is being adopted and how its use is affecting workers across industries, wage categories, and sociodemographic groups. Relatedly, research on the trajectory of AI’s use and impact in different industrial sectors may be useful in the FoW initiative, prioritizing the development of future research, OSH, and skills-building initiatives.

Current and future efforts to understand the impact of AI on equity outcomes are further complicated by the limited equity-related variables in current OSH data collection systems [119]. For example, Occupational Safety and Health Administration illness and injury logs, Bureau of Labor Statistics’ Survey of Occupational Injuries and Illnesses, workers’ compensation data systems do not collect race/ethnicity data, and other equity variables (e.g., education, income level, geographic location, etc.) may be limited [120]. Several ways have been proposed to address this, including: (1) collecting race/ethnicity data in these systems; (2) linking OSH data to systems that do include race/ethnicity; and (3) using algorithms to predict race/ethnicity from other available data points, such as name and address [121]. As with any data related to demographic characteristics, privacy concerns and potential misuse based on social bias are of concern. Conversely, AI has recently proven useful during the COVID-19 pandemic to overcome limitations of public health data collection systems with respect to workers from historically underrepresented groups. Specifically, a recent study used artificial intelligence to analyze news reports of COVID-19 workplace outbreaks to identify social factors that are often not fully captured in public health data systems (such as race, ethnicity, and nativity) that could potentially have affected disease transmission [122].

### 4.2. Considerations for OSH Professionals

While AI’s use in the workplace is inescapable, OSH professionals must consider how the development and use of these tools impact equity. Accordingly, assembling a diverse and multidisciplinary team of collaborators and including workers in the process may promote OSH equity in the development and implementation of AI tools in the workplace. A social approach to addressing the equity impacts of AI requires a paradigm shift for OSH professionals [117]. Understanding how social structures circumscribe the development, implementation, and impact of AI in the workplace is an essential first step to ensuring equitable distribution of the benefits of AI in the workforce and realizing its potential to reduce OSH inequities. Understanding the complexities of AI will assist OSH professionals in the protection of worker wellbeing and the promotion of health equity. This may require the training of OSH professionals to be better prepared to support workers in the FoW.

An OSH approach that anticipates trends using strategic foresight or proactive intervention can help to mitigate the negative effects of AI on OSH equity [38,45,60,84]. OSH professionals might consider the effects of AI integration on workers’ lived experiences and on low-resourced communities and organizations, including an understanding of how asymmetrical power relationships along axes such as race, ethnicity, sex, gender, nativity, and class impact the distribution of work-related benefits and risks of AI.

### 4.3. Limitations

This study was a scoping review; therefore, the focus was on the breadth rather than the depth of information. As a result, an assessment of the methodological limitations and risk of bias in the evidence was not performed. Inclusion criteria around date of publication (i.e., 2005–2021) may have limited our search results. The sample largely consisted of U.S.-based literature, limiting our ability to perform a comparison with the international context of AI and OSH equity. International comparison of OSH equity is, however, quite difficult due to the diversity of labor markets, employment, and working conditions globally [123,124].

There is limited research on AI that has explicitly studied or mentioned issues of health equity. Although our search uncovered three case studies, the articles included in the review were largely theoretical or conceptual as opposed to applied research [97,125,126]. The empirical studies relied on large datasets or policy analysis to make claims about AI, OSH, and equity in the FoW. This lack of applied research limits our understanding of the practical outcomes and limitations of AI. Relatedly, qualitative research was significantly lacking, which further limits our understanding of workers’ lived experience.

## 5. Conclusions

AI implementation is already pervasive in many industries, and its use is rapidly expanding [127,128]. AI systems are tools used by employers primarily to make workplaces more efficient and effective. For example, in healthcare AI is used to improve diagnoses and treatment precision [96], while in business AI tools track productivity to maximize profit [56]. While research is limited, study team assumptions related to AI as a potential contributor to health disparities (i.e., unemployment, wages, etc.) were confirmed by the literature.

As with any tool, AI’s impact depends on how it is used, and whether it reduces or exacerbates inequities is determined by its development, application, and evaluation. While AI may lead to safer workplaces through the use of assistive technology, concerns around bias in AI programming, recruitment and hiring, job insecurity and unemployment, and personal data use demonstrate a need to further explore the intermediate influence mechanisms of AI on workforce health and equity. To that end, this scoping literature review recognizes that significant changes in the FoW are inevitable, and as such begins to elucidate barriers and facilitators of AI in promoting OSH equity. AI’s role in OSH equity is vastly understudied. There is an urgent need for multidisciplinary research that addresses where and how AI is being adopted and evaluated and how its use is affecting OSH across industries, wage categories, and sociodemographic groups. OSH professionals can play a significant role in identifying strategies to ensure the benefits of AI in promoting workforce health and wellbeing are equitably distributed.

The findings and conclusions in this report are those of the authors, and do not necessarily represent the official position of the National Institute for Occupational Safety and Health or Centers for Disease Control and Prevention.

## Figures and Tables

**Figure 1 ijerph-20-06221-f001:**
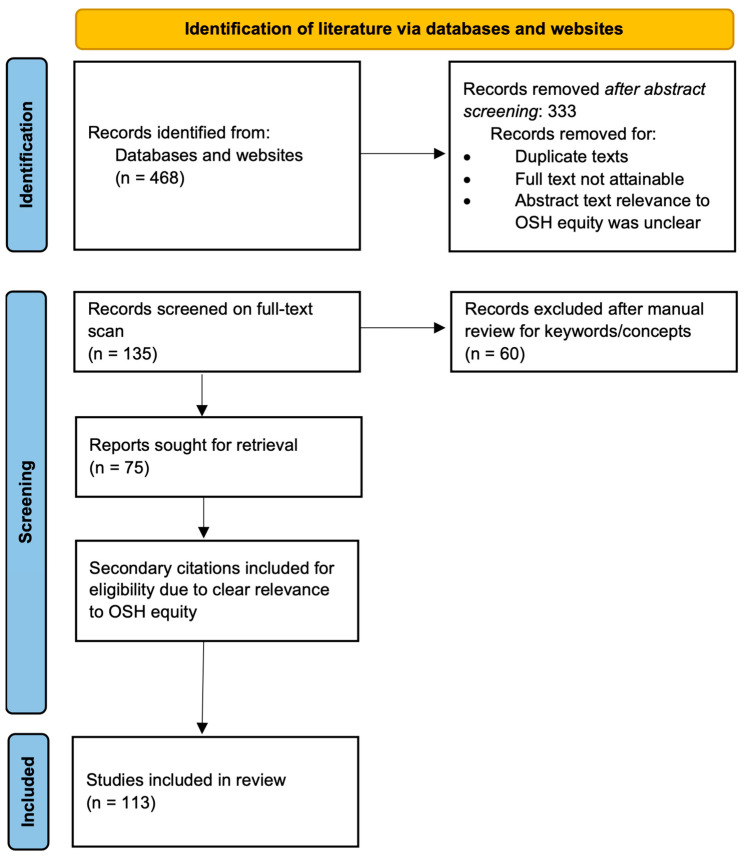
This adapted version of the PRISMA guidelines demonstrates the scoping review search strategy used in this review in order to illustrate the process used and the resulting articles.

**Table 1 ijerph-20-06221-t001:** Literature search keywords and concepts.

Keyword/Concept	Definition
Human-Centered Systems Design	The AI is designed, preferably using a multidisciplinary approach, with humans in mind, considering significant users (workers) or groups of users who are expected to interface or take advantage of the AI [25,26].
Privacy	Employees understand the type of information that is kept private and who maintains control over this information. This includes personal identifiable data and information used by or produced by AI systems.
Confidentiality	Employers manage employee occupational and personal data in an arranged and trustworthy manner.
Bias	While there are many forms of bias, conceptually they all meet the following International Organization for Standardizations’ definition: “the degree to which a reference value deviates from the truth” [27].
Job Security ^1^	An employee’s expectation that AI integration will not disrupt continuity in their current job or create concern about the future permanence of a job.

^1^ Job Security was identified as an emergent keyword in early search stages due to its prominence in the literature.

**Table 2 ijerph-20-06221-t002:** Search sources.

Databases
ACM Digital Library
Business Source Complete
Google Scholar
NexisUni
PsycINFO
Public Affairs Information Service International Index
PubMed
Roper Center for Public Opinion Research
Scopus
Web of Science
Top 50 Think Tanks ^1^
Center for Strategic and International Studies
Carnegie Endowment for International Peace
Heritage Foundation
Peterson Institute for International Economics
Urban Institute
Woodrow Wilson International Center for Scholars
Center for American Progress
Atlantic Council
Hudson Institute
Council on Foreign Relations
Belfer Center for Science and International Affairs
Cato Institute
Center for New American Security
American Enterprise Institute for Public Policy Research
James A. Baker III Institute for Public Policy
Stimson Center
Human Rights Watch
Resources for the Future
Freedom House
German Marshall Fund of the United States
Hoover Institution
World Resources Institute
McKinsey Global Institute
National Bureau of Economic Research
Inter-American Dialogue
Information Technology and Innovation Foundation
Center for Global Development
United States Institute of Peace
Acton Institute for Study of Religion and Liberty
Pew Research Center
Center on Budget and Policy Priorities
Manhattan Institute for Policy Research
Economic Policy Institute
National Bureau of Asian Research
Worldwatch Institute
Mercatus Center
Carnegie Council for Ethics in International Affairs
Migration Policy Institute
Independent Institute
Center for Climate and Energy Solutions
Middle East Institute
Earth Institute
Center for National Interest
Bipartisan Policy Center
Aspen Institute
EastWest Institute
New America Foundation
Center for Transatlantic Relations
U.S. Government Agencies
Congressional Research Service
Department of Defense
Department of Homeland Security
Department of Health and Human Services
Centers for Disease Control and Prevention
Department of Labor

^1^ Presented in order of appearance, as represented by the University of Pennsylvania Think Tank Index Report [29].

**Table 3 ijerph-20-06221-t003:** Scoping review search terms.

Artificial Intelligence	Occupational Safety and Health	Equity
“artificial intelligence”	“occupational safety”	equity
AI	“occupational health”	equitable
algorithm *	“industrial hygiene”	inequity
automat *	“worker safety”	inequitable
robot *	“workplace safety”	justice
bots	“safe workplace*”	disparit *
“reactive machine *”	“workplace health”	
“limited memory”	“worker protection*”	
“theory of mind”		

An asterisk is used to broaden the search to include all possible word endings.

**Table 4 ijerph-20-06221-t004:** Frequency of Keywords/Concepts by Source Type.

Keyword/Concept	Number of Articles (N = 113) n (%)
Job Security	58 (51%)
Peer reviewed	20
White	21
Grey	17
Human-Centered Systems Design	48 (42%)
Peer reviewed	20
White	15
Grey	13
Privacy	43 (38%)
Peer reviewed	15
White	17
Grey	11
Bias	40 (35%)
Peer reviewed	13
White	16
Grey	11
Confidentiality	39 (35%)
Peer reviewed	13
White	15
Grey	11

## Data Availability

Data sharing not applicable.

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
