# Peer review of "Occupational Safety and Health Equity Impacts of Artificial Intelligence: A Scoping Review"

_ijerph, 2023, doi:10.3390/ijerph20136221_

Round 1

Reviewer 1 Report

1. This manuscript is well-written, clear and a valuable review of literature. However, overall the tone is very US-focused. Nothing wrong with that, but it should be made clearer that many of the emerging messages are derived from assessment of US evidence and that conclusions may  apply predominantly to US practices.

2. While international peer-reviewed literature will have been included in the bibliometric databases surveyed, grey literature perhaps would not have been. Therefore, given the emphasis by the authors in accessing grey literature from US-based government departments and think tanks perhaps there is a resultant evidence bias against international grey literature?

3. Even when cited (e.g. reference 118 on EU-OSHA), the non-US literature is not much discussed and this seems a missed opportunity to put some of the specific points (e.g. effect of AI on different industries, hiring practices) into their international context. This could have been particularly pertinent for comparing the US with the European situation where forthcoming legislation in the AI Act is designed to manage key issues described by the authors e.g. AI, algorithmic bias and hiring decisions. See, for example, Kullmann in Global Workplace Law and Privacy 24 January 2022 The Interconnection between the AI Act and the EU’s Occupational Safety and Health Legal Framework - Global Workplace Law & Policy (kluwerlawonline.com). 

4. This limitation of the manuscript in terms of international discussion could be mentioned in section 4.8.

5. In the Discussion, particularly in section 4.3, it would also be helpful to the reader to have more assessment of likely timescales for impacts on OSH equity. For example, prediction of differential impacts on different occupations, because there may be implications for prioritising the solutions on the basis of best practice and for conducting new research.

6. Table 3 - did the authors consider including the term "machine learning"?

7. Section 4.5 "Summative interpretation" - this section seems mainly to duplicate the points already clearly made - is it needed?

8. Suggest that the authors also cross refer to the recent (2022) IJERPH special issue "Frontiers in occupational health and safety management", especially the editorial by Ramos et al, which refers to AI.

Author Response

Dear Reviewer 1,

Thank you for your initial review of our paper, “Occupational Safety and Health Equity Impacts of Artificial Intelligence in the Future of Work: A Scoping Review.” We have revised our original submission in response to your comments.

Reviewer 1:

Reviewer Comments

Response to Reviewer

1

This manuscript is well-written, clear and a valuable review of literature. However, overall the tone is very US-focused. Nothing wrong with that, but it should be made clearer that many of the emerging messages are derived from assessment of US evidence and that conclusions may  apply predominantly to US practices.

Thank you for your insightful comment. We have revised the language so that it is clear that the evidence is primarily from the U.S., and therefore applies to U.S. practices.

2

While international peer-reviewed literature will have been included in the bibliometric databases surveyed, grey literature perhaps would not have been. Therefore, given the emphasis by the authors in accessing grey literature from US-based government departments and think tanks perhaps there is a resultant evidence bias against international grey literature?

In our methods section, we clarified the sources used in the scoping review, particularly in the collection of grey literature (i.e., governmental literature from the U.S. and development of the international University of Pennsylvania Think Tank Index Report)

We have also noted that the focus of the scoping review on a U.S. context, was a limitation of the study.

3

In the Discussion, particularly in section 4.3, it would also be helpful to the reader to have more assessment of likely timescales for impacts on OSH equity. For example, prediction of differential impacts on different occupations, because there may be implications for prioritising the solutions on the basis of best practice and for conducting new research.

Beyond the information previously included in the manuscript, the authors do not have any additions related to the timescales of AI’s impact on different industrial sectors. We agree with your comment and have added this as an area for future research in Lines 411-414.

4

Table 3 - did the authors consider including the term "machine learning"?

“Machine learning” was not included as a search term. The study team chose to use the search terms “algorithm” “reactive machine” and “limited memory” as more specific terminology that refers to machine learning models.

5

Suggest that the authors also cross refer to the recent (2022) IJERPH special issue "Frontiers in occupational health and safety management", especially the editorial by Ramos et al, which refers to AI.

We appreciate the suggestion to review this article. We have cross referenced Ramos et.al., in Lines 44-45.

We appreciate your thorough review of our manuscript, and we look forward to hearing back from you.

Sincerely,

Elizabeth Fisher

Reviewer 2 Report

Dear Authors,

I read your paper on the role of AI in the future of work. In general, we may say that cooperation with AI devices and programs is inevitable and in every field of work some parts of AI are present. I like your research problems because you address the issue of equity and your research are reliable and unbiased. AI is not clearly positive nor negative, its efficiency depends on a human who programs and uses it. That is why, the effects of its usage are both positive and negative as it can be seen i.e. in recruiting process. In this case the most important is human dignity which can be easily violated because of replacing a human recruiter with AI procedures.

To make your article even better, I would recommend extension of conclusions and showing the criteria of inclusion for the articles chosen to analysis because they are not clear to the reader. In the future, when you do similar research, I advise that you use some tools which allow you provide not only qualitative but also quantitative analysis.

To sum up, I consider your article as very good and recommend it to be publish.

Author Response

Dear Reviewer 2,

Thank you for your initial review of our paper, “Occupational Safety and Health Equity Impacts of Artificial Intelligence in the Future of Work: A Scoping Review.” We have revised our original submission as track-changes in response to your comments.

Reviewer 2:

Reviewer Comments

Response to Reviewers

1

I would recommend extension of conclusions and showing the criteria of inclusion for the articles chosen to analysis because they are not clear to the reader.

We have expanded the conclusion and included a PRISMA to better demonstrate the inclusion criteria. 

2

In the future, when you do similar research, I advise that you use some tools which allow you provide not only qualitative but also quantitative analysis.

Thank you for your suggestion. We agree that future studies could include more in-depth quantitative analysis given the proper data analysis tools.

We appreciate your thorough review of our manuscript, and we look forward to hearing back from you.

Sincerely,

Elizabeth Fisher

Reviewer 3 Report

Thanks to authors for a very good review paper. My suggestions are as follows:

1. Introduction

Understanding the impact of AI is very important, authors are suggested to  show more background on AI' impact on occupational health, especially from typical studies around the world.

2. Materials and Methods

Authors have obtained a lot of  research materials from different databases. I think Fig1 does't contain necessary informations of the methods.

3. Results

It seems only a summary of the frequency analysis of words. In terms of the structure of the manuscript, it is not recommended as a separate chapter.

4. Discussion

Whether  the previous questions in part 2 are better answered in the discussions. I don't think so.

 There is a good limitations writing, and I suggested that AI impact on health equity may be a influence mechanism, which may contain different transmission mechanisms, and more should be done to discover the intermediate influence factors.

The paper gives out an important humanistic direction for AI research, and is a particularly good content. 

Thanks all authors for their contributions.

Author Response

Dear Reviewer 3,

Thank you for your initial review of our paper, “Occupational Safety and Health Equity Impacts of Artificial Intelligence in the Future of Work: A Scoping Review.” We have revised our original submission as track-changes in response to your comments.

Reviewer 3:

Reviewer Comments

Response to Reviewers

1

Understanding the impact of AI is very important, authors are suggested to show more background on AI' impact on occupational health, especially from typical studies around the world.

The introduction has been expanded to include additional background on AI’s use in workplaces. Please see Lines 36-51.

2

There is a good limitations writing, and I suggested that AI impact on health equity may be a influence mechanism, which may contain different transmission mechanisms, and more should be done to discover the intermediate influence factors.

.

Thank you for your comment and we agree that AI is an influence mechanism on occupational safety and health equity, with a variety of transmission mechanisms. In the conclusion, which has been expanded, we have added a recommendation to further explore the intermediate influence mechanisms of AI on occupational safety and health equity.

We appreciate your thorough review of our manuscript, and we look forward to hearing back from you.

Sincerely,

Elizabeth Fisher

Reviewer 4 Report

The title is in the Future of Work but many AI tools are using now, is that an appropriate word?

Lines 10-15 can be condensed.

What is health equity? Should that language barrier affect the equity? Tech and wealth level as a kind of barrier? If that and this have not been stated, these should be stated in limitation.

Research gap should be stated in the paper.

scoping review should be clearly defined too.

The research method should be stated in abstract.

Lines 21-23, this is not needed “Conclusion: There is a need for a multidisciplinary research agenda to study the role of AI in mitigating OSH inequities and identify strategies that ensure that benefits of AI in promoting workforce health and wellbeing are equitably distributed.” Please state the academic, practical and policy contributions.

Why online databases and on the websites of the Top 50 think tanks, as represented by the University of Pennsylvania are selected? Is there a bias?

Figure 1 should be replaced by PRISMA approach, state the inclusion and exclusion criteria clearly https://pubmed.ncbi.nlm.nih.gov/36117599/

Why and how 468 articles are identified? What are the keywords used? All these need to be stated in form of PRISMA approach (please also define what PRISMA is in the paper).

Citations like [20,27,33,57,97], should state what these stated in the paper.

There are a lot of papers about AI and health and safety but have not been included.

As such, how do the authors select the papers?

Lines 324-325, Education systems can be adapted so that all communities are prepared for the future of work, say, at least how to use some tech should be included.

skills gap should include digital divide as a concept.

Table A1, Lead article vs. secondary citation is not needed.

Table A1, Key words should keywords

Conclusion has to be expanded.

Too many websites in the references should be replaced by top journal articles.

In health and safety, there are a lot of top journals like safety science and others but the authors do not have covered that.

Author Response

Dear Reviewer 4,

Thank you for your initial review of our paper, “Occupational Safety and Health Equity Impacts of Artificial Intelligence in the Future of Work: A Scoping Review.” We have revised our original submission as track-changes in response to your comments.

Reviewer 4:

Reviewer Comments

Response to Reviewers

1

The title is in the Future of Work but many AI tools are using now, is that an appropriate word?

To clarify the title of our paper, we have provided more information on the U.S. National Institute for Occupational Safety and Health Future of Work Initiative, which also further describes the research gap the authors aimed to explore.

2

Abstract: Lines 10-15 can be condensed. Lines 21-23, this is not needed

The research method should be stated in abstract.

Conclusion: There is a need for a multidisciplinary research agenda to study the role of AI in mitigating OSH inequities and identify strategies that ensure that benefits of AI in promoting workforce health and wellbeing are equitably distributed.” Please state the academic, practical and policy contributions.

Thank you for your recommendations to improve the quality of the abstract. Accordingly, we have condensed Lines 10-15, and removed former Lines 21-23.

Scoping review is stated in Line 14 of the abstract.

We have revised the conclusion to state the academic and practical findings from the scoping review.

3

What is health equity? Should that language barrier affect the equity? Tech and wealth level as a kind of barrier? If that and this have not been stated, these should be stated in limitation.

Occupational safety and health inequities are defined n Lines 54-62 and includes economic disadvantage (or wealth level) and access to resources/technologies. ‘Language barriers’ has been included as an additional contributor to occupational safety and health inequity in Line 59.

4

Research gap should be stated in the paper.

The research gap has been more clearly stated in the paper in Lines 81-84.

5

Scoping review should be clearly defined too.

Scoping review was defined in the methods in Lines 87-90.

6

Why online databases and on the websites of the Top 50 think tanks, as represented by the University of Pennsylvania are selected? Is there a bias?

In our methods section, we clarified the sources used in the scoping review, particularly in describing the University of Pennsylvania Think Tank Index Report, which considers over 10,000 international think tanks in the development of their international survey. 

7

Citations like [20,27,33,57,97], should state what these stated in the paper.

Thank you for your comment. Lengthy in-text citation lists were broken up to better reflect the content cited where appropriate.

8

There are a lot of papers about AI and health and safety but have not been included. As such, how do the authors select the papers?

Please note that the inclusion criteria and search process was clarified in the PRISMA. We have added peer-reviewed literature where appropriate. While there is a growing body of literature around AI and occupational safety and health, few address issues of health equity specifically.

9

Lines 324-325, Education systems can be adapted so that all communities are prepared for the future of work, say, at least how to use some tech should be included. Skills gap should include digital divide as a concept.

Thank you for the suggestion. The statement has been revised in Lines 350-351.

10

Table A1, Lead article vs. secondary citation is not needed.

We appreciate the recommendation. However, we have decided to keep the lead article vs. secondary citation in Table A1 as further information related to inclusion criteria per the request of other reviewers.

11

Table A1, Key words should keywords

We have revised the spelling of keywords in Table A1. We have also run the revised version of our manuscript through an automated editing service to ensure the appropriate use of English language and grammar.

12

Too many websites in the references should be replaced by top journal articles. In health and safety, there are a lot of top journals like safety science and others but the authors do not have covered that.

Scoping review was defined in the methods, and we request that the reviewers note that scoping reviews are designed to include topical/popular literature, which accounts for the number of websites listed as citations. We appreciate your recommendation and have added citations from peer-reviewed sources where appropriate.

We appreciate your thorough review of our manuscript, and we look forward to hearing back from you.

Sincerely,

Elizabeth Fisher

Round 2

Reviewer 4 Report

1.     Many of the previous comments were ignored, please revise.

2.     Title is problematic, what is a scoping review? Why could you use present data to study the impacts in the future work? The first two lines are unclear.

3.     Title is scoping review but the method is PRISMA?

4.     Line 14, we won’t write Methods: A scoping review was completed to summarize, there should not be one word only before colon.

5.     The authors may need to simplify the abstract part, some of the details in the research may not be revealed here.

6.     The importance of this research may be added in the abstract part to make this part more attractive.

7.     The authors may need to point out the research gaps based on the past studies and indicate what gaps will be filled by this research in the introduction part.

8.     A brief introduction of the coming sections (the structure of this paper) should be listed at the end of the introduction part.

9.     There should be a section with title “Literature review” after the introduction part, and some of the content in the introduction part could be directly moved to the LR part, the authors may rise a discussion based on the past studies in this section about the gaps, importance, hypothesis., etc.

10.    Line 113, Through an iterative process? Why? What is that process?

11.    Separate the academic journal results from think tank etc. You should not mix them together.

12.    The research questions may not be revealed until the materials and methods section, please state the research questions in the introduction part, and a literature review is needed.

13.    What are gaps in addressing AI and OSH equity? Please provide a link for all these paper.

What are key issues for OSH/Industrial Hygiene professionals to understand around AI?” what are these lines? The authors may need to revise the format of this content. A more detail discussion about OSH for AI https://www.sciencedirect.com/science/article/abs/pii/S0925753521004422 is needed before we enter to equity. There should be a sequent in discussion

8. The authors may need to make a detailed introduction of the research methods.

9. The authors may need to make a brief introduction of section 3.4, the summary of this section did not mention this section.

10. Some of the references may seem a bit outdated, the authors may check them.

11. The academic or policy contribution may be mentioned in the conclusion part.

12. Please remove as much websites as possible from references

Author Response

Dear Reviewer 4,

Thank you for your  review of our paper, “Occupational Safety and Health Equity Impacts of Artificial Intelligence: A Scoping Review.” We have revised our previous submission in response to your comments, and list your comment, and our response in the table below:

1.     Many of the previous comments were ignored, please revise.

We thank the reviewer for their thoughtful feedback. Where possible, the authors have considered the suggested changes and updated the manuscript accordingly.  The authors have done their best in responding to the changes which are appropriate for the scoping review performed.

2.     Title is problematic, what is a scoping review? Why could you use present data to study the impacts in the future work? The first two lines are unclear.

Thank you for the comment.

The authors have updated the title to “Occupational Safety and Health Equity Impacts of Artificial Intelligence: A Scoping Review”.   Hopefully this improves clarity for the reader.

3.     Title is scoping review but the method is PRISMA?

This study was a scoping review, defined as a literature review to synthesize the research in a topical area to serve as a starting point for future research. As requested by the reviewer, the authors described how the method used incorporated principles associated with PRISMA, as described in lines 108-109. 

4.     Line 14, we won’t write Methods: A scoping review was completed to summarize, there should not be one word only before colon.

The authors adjusted the text accordingly.

5.     The authors may need to simplify the abstract part, some of the details in the research may not be revealed here.

Adjustments were made to simplify, while considering comments from other reviewers.  

6.     The importance of this research may be added in the abstract part to make this part more attractive.

The authors considered potential adjustments and comments from other reviewers, while simplifying the abstract.

7.     The authors may need to point out the research gaps based on the past studies and indicate what gaps will be filled by this research in the introduction part.

Authors have clearly identified the research gap to be filled by a scoping review with reference to the FoW initiative, noting the importance of this effort due to limited literature available.

8.     A brief introduction of the coming sections (the structure of this paper) should be listed at the end of the introduction part.

The authors revised the sentence on line 82 to read, “A scoping review of the literature, thematic representation of findings, as well as a discussion of considerations for OSH professionals to address AI and OSH equity are included.”

9.     There should be a section with title “Literature review” after the introduction part, and some of the content in the introduction part could be directly moved to the LR part, the authors may rise a discussion based on the past studies in this section about the gaps, importance, hypothesis., etc.

The authors believe this paper is a scoping review and adding a literature review section to the introduction is not appropriate.

10.    Line 113, Through an iterative process? Why? What is that process?

Additional text was added to explain how an interative approach was used for this review.  

11.    Separate the academic journal results from think tank etc. You should not mix them together.

Additional clarification between peer-reviewed, white, and grey literature was added.  

12.    The research questions may not be revealed until the materials and methods section, please state the research questions in the introduction part, and a literature review is needed.

A scoping review identifies gaps in research.   Our research questions were used to guide our selection of literature, and should be included in the methods used for the scoping review.  Lines 78-80 articulate the purpose and need for the literature review.

13.    “● What are gaps in addressing AI and OSH equity? Please provide a link for all these paper.

● What are key issues for OSH/Industrial Hygiene professionals to understand around AI?” what are these lines? The authors may need to revise the format of this content. A more detail discussion about OSH for AI https://www.sciencedirect.com/science/article/abs/pii/S0925753521004422is needed before we enter to equity. There should be a sequent in discussion

While the scoping review identified opportunities for future research, the literature is limited and was not adequate to identify all gaps and address all issues.

8. The authors may need to make a detailed introduction of the research methods.

While the authors do not fully understand the recommendation provided by the reviewer, language was added to lines 87-90 to best fulfill this recommendation.

9. The authors may need to make a brief introduction of section 3.4, the summary of this section did not mention this section.

The authors do not fully understand the recommendation provided by the reviewer.

10. Some of the references may seem a bit outdated, the authors may check them.

This is a scoping review limited to the years 2005-2021 and our findings reflect literature from this time period.

11. The academic or policy contribution may be mentioned in the conclusion part.

Noting this is a scoping review, the authors are focused on and limited conclusions to research gaps.

12. Please remove as much websites as possible from references

The scoping review was defined in the methods, and which are designed to include topical/popular literature, which accounts for the number of websites listed as citations. We appreciate your recommendation and have added citations from peer-reviewed sources where appropriate.

We appreciate your assistance in improving the quality and clarity of our manuscript.   The updated manuscript has been attached for your review.

Thank you.

Jay Vietas
